# Assessment of Pain-Related Fear in Indigenous Australian Populations Using the Fear of Pain Questionnaire-9 (FPQ-9)

**DOI:** 10.3390/ijerph19106256

**Published:** 2022-05-20

**Authors:** Manasi Murthy Mittinty, Pedro H. R. Santiago, Lisa Jamieson

**Affiliations:** 1Faculty of Medicine and Health, The University of Sydney, St Leonards, NSW 2050, Australia; 2Indigenous Oral Health Unit, Faculty of Health and Medical Sciences, The University of Adelaide, Adelaide, SA 5000, Australia; pedro.ribeirosantiago@adelaide.edu.au (P.H.R.S.); lisa.jamieson@adelaide.edu.au (L.J.)

**Keywords:** Aboriginal Health, fear of pain, disparity, chronic pain, rural area

## Abstract

In this study, we examined the psychometric properties of the Fear of Pain Questionnaire (FPQ-9) in Indigenous Australian people. FPQ-9, a shorter version of the original Fear of Pain Questionnaire-III, was developed to support the demand for more concise scales with faster administration time in the clinical and research setting. The psychometric properties of FPQ-9 in Indigenous Australian participants (*n* = 735) were evaluated with network psychometrics, such as dimensionality, model fit, internal consistency and reliability, measurement invariance, and criterion validity. Our findings indicated that the original FPQ-9 three-factor structure had a poor fit and did not adequately capture pain-related fear in Indigenous Australian people. On removal of two cross-loading items, an adapted version Indigenous Australian Fear of Pain Questionnaire-7 (IA-FPQ-7) displayed good fit and construct validity and reliability for assessing fear of pain in a sample of Indigenous Australian people. The IA-FPQ-7 scale could be used to better understand the role and impact of fear of pain in Indigenous Australian people living with chronic pain. This could allow for more tailored and timely interventions for managing pain in Indigenous Australian communities.

## 1. Introduction

Fear is a biological response to pain; however, heightened fear of pain may become maladaptive [1] in nature and has been implicated in adverse physical and psychological sequelae to pain, including disability, anxiety, and depression. Heightened fear of pain tends to perpetuate hypervigilance to bodily sensations, reinforcing inactivity and thereby delayed recovery [2,3]. On contrary, lower fear of pain propagates positive engagement in physical activity, thereby promoting recovery [4,5]. Effective assessment for identifying patients at greater risk of pain-related fear can be an important way to identify patients who may also be at greater risk of adverse outcomes, creating an opportunity to intervene.

The Fear of Pain Questionnaire-III (FPQ-III) [6] developed by McNeil and Rainwater is a self-report measure that assesses fear associated with stimuli known to elicit pain. The original FPQ-III used 30 items to assess three subscales: Fear of Medical/Dental Pain, Fear of Minor Pain, and Fear of Major Pain. Application of the FPQ-III has been validated and tested in both clinical and non-clinical samples [7,8,9] and has been adapted in languages other than English [4,10,11,12,13]. A shorter version of the FPQ-III, the Fear of Pain Questionnaire-9 (FPQ-9) [14], was recently developed to support the demand for concise scales with faster administration time. Although the shorter version has only nine items as compared to thirty items of FPQ-III, it has been found to be a good alternative to the original scale [14,15]. The FPQ-9 is also designed to assess similar three dimensions of painful stimulus, namely the Fear of Medical/Dental Pain, Fear of Minor Pain, and Fear of Severe Pain. Previous studies that examined the FPQ-9 psychometric properties indicated good model fit of the three-factor structure [4,14,15], adequate internal consistency (ranging from 0.72 to 0.84) and test-retest reliability (ranging from 0.74 to 0.87) [14,15], and good convergent/discriminant validity [7].

Considering the crucial role of fear of pain in optimizing recovery, it is important to assess its function in Aboriginal and Torres Strait Islander people of Australia, herein respectfully referred to as Indigenous Australian people. Given the long history of trauma related to loss of land and culture, racism, and abuse, there remains a significant disparity in understanding how pain and its corollaries are measured and reported, often without accommodating to Indigenous Australian cultural sensitivities [16], which is a sufficient cause for clinical quality and safety concerns. It remains to be discovered if pain-related fear shows identical patterns and impact in Indigenous communities and whether it could be used for improving patient care. These disparities primarily exist due to the absence of scales developed, adapted, and/or validated for application in Indigenous Australian people.

Relief from pain is a global right, and its optimal assessment is the first step towards improving pain care for Indigenous communities. Therefore, this study aims to examine the psychometric properties, including the suggested three-factor structure, utility, and validity of the FPQ-9 in a clinical sample of Indigenous Australian population.

## 2. Materials and Methods

### 2.1. Design and Setting

Participants were recruited as a part of a larger clinical study on human papilloma virus (HPV) and its impact in Indigenous Australian communities, with baseline data collection between February 2018 and January 2019 and 12-month follow-up between March 2019 and March 2020 [17]. The HPV study inclusion criteria were identification as Aboriginal and/or Torres Strait Islander and being aged 18 years or older. The study recruitment strategies included service agreements with Aboriginal community-controlled health organizations in South Australia, collaboration with Aboriginal community elders, and encouraging word of mouth. The HPV study received oversight from a senior Indigenous project manager and employed three Indigenous staff (who received extensive training in research and data collection). A sample of 1011 Aboriginal adults was recruited at baseline, and 750 participated in the 12-month follow-up (74% retention rate; the study was suspended early due to the COVID-19 pandemic). All recruitment and data collection procedures were performed following the ethical standards laid down by the 1964 Declaration of Helsinki and its later amendments. Ethics approval was obtained from the University of Adelaide Human Research Ethics Committee (H-2016–246) and the Aboriginal Health Council of South Australia (04–17-729). For their time each, participant received reimbursement of AUD 50 voucher at baseline, 12-month, and 24-month follow-ups.

### 2.2. Measures

Sociodemographic information, such as participants’ gender, age, education level, marital status, and employment status were collected along with clinical information, such as participants’ medical history, particularly years since diagnosis of chronic painful condition(s). In line with the updated definition, pain that persisted beyond its usual healing time capped at 3 months was defined as chronic pain [18]. Information about participants’ average pain intensity was recorded on 0 to 10 numerical rating scale, with 0 being absence of pain to 10 being the worse pain possible.

#### 2.2.1. EQ-5D-5L

The EQ-5D-5L [19] subscales were used to measure change in quality of life due to pain/discomfort (1 = “I have no pain or discomfort”, 2 = “I have slight pain or discomfort”, 3 = “I have moderate pain or discomfort”, 4 = “I have severe pain or discomfort”, and 5 = “I have extreme pain or discomfort”) and change in quality of life due to anxiety/depression (1 = “I am not anxious or depressed”, 2 = “I am slightly anxious or depressed”, 3 = “I am moderately anxious or depressed”, 4 = “I am severely anxious or depressed”, and 5 = “I am extremely anxious or depressed”). The EQ-5D-5L has been previously validated for Aboriginal Australians [17].

#### 2.2.2. Fear of Pain

The FPQ-9 [14], a self-report 9-item measure, was administered to assesses fear and anxiety related to potential painful stimulus. Each item is rated on 5-point Likert scale, with total score ranging from 9 to 45. Following recommendations for cultural adaptations of psychological instruments [20], the FPQ-9 was evaluated by an Indigenous Reference Group, established to provide governance and research oversight [17], prior to the inclusion of the instrument in the current study. The Indigenous Reference Group endorsed the inclusion of the FPQ-9 in the study and its application among Indigenous Australians, recommending further investigation of its psychometric properties.

### 2.3. Statistical Analysis

Percentages, means, and standard deviations were computed to describe the sociodemographic and clinical variables of the participants. The statistical analyses were conducted with R packages EGAnet [21], powerly [22], and semTools [23]. Missing values in the original sample (*n* = 750) for individual items ranged from 0.4% to 1.1%, so missingness was unsubstantial, and multiple imputation was not required [24]. All analysis was conducted with complete cases (*n* = 735). A multi-method approach was applied to comprehensively evaluate the psychometric properties.

#### 2.3.1. Power Calculation

The Monte Carlo simulation method was used to calculate the sample size needed for estimating the network model with a specified power of 80%. The sample size was calculated to estimate a GGM with 9 nodes (the 9 nodes of the FPQ-9 network) and edge density of 0.3 reaching a sensitivity of 60% regarding true estimated edges across 80% of all cases.

#### 2.3.2. Network Estimation

The Gaussian Graphical Model (GGM) was used, wherein nodes indicate items and edges indicate partial correlation coefficients between items. Since the Fear of Pain Questionnaire is composed of ordinal polytomous items, we used as input polychoric correlation coefficients and estimated the network with the Graphical LASSO (Least Absolute Shrinkage and Selection Operator) [25]. To provide a visual representation of the network, we plotted the network models with the Fruchterman–Reingold algorithm.

#### 2.3.3. Exploratory Graph Analysis (EGA)

The EGA was performed to determine the dimensional structure of FPQ-9 for Indigenous Australian population using the Louvain algorithm [26,27]. EGA identified the dimensionality in the development sample (*n* = 220), and evaluation of model fit and all subsequent analysis (i.e., measurement invariance) were conducted in the validation sample (*n* = 515).

#### 2.3.4. Model Fit

Model fit was evaluated with the Total Entropy Fit Index using Von Neumman entropy (TEFIvn). Lower values of the TEFIvn indicate better fit. We also employed traditional Confirmatory Factor Analysis (CFA) fit indices, such as the Root Mean Squared Error of Approximation (RMSEA) and Comparative Fit Index (CFI). We estimated factor models with weighted least squares and a mean- and variance-adjusted (WLSMV) test statistic. To evaluate RMSEA and CFI, we followed the latest recommendations in factor analytical research that CFI ≥ 0.900 indicates acceptable fit, while RMSEA ≤ 0.050 indicates good fit, and RMSEA ≥ 0.100 indicates unacceptable fit [28]. We then evaluated the original three-factor FPQ-9 structure and the EGA-identified structure. We also conducted model re-specifications based on modification indices (MI) and the standardized expected parameter change (SEPC).

#### 2.3.5. Internal Consistency and Reliability

Internal consistency reliability was calculated with McDonald’s coefficient Ω [29]. We also evaluated corrected item-total correlations (CITC) with non-parametric correlation Kendall’s *τ*.

#### 2.3.6. Measurement Invariance

After a model with good fit was established, we proceeded to evaluate measurement invariance by sex and age. To do so, we followed recent recommendations from Svetina, Rutkowski, and Rutkowski [30]. Initially, we fitted a Multigroup CFA in both groups (i.e., men and women) to investigate configural invariance and whether the same items measure the same factors across both groups. To evaluate configural invariance, the χ^2^, CFI, and RMSEA and their previously described cut-off points were used. All multigroup CFA models were estimated with theta parameterization, and model identification followed Wu and Estabrook [31]. After the establishment of configural invariance, we then progressively constrained to be equal across groups: (1) thresholds; (2) factor loadings (metric invariance); and (3) intercepts (scalar invariance). Since scalar, metric, and configural models are nested, to evaluate metric and scalar invariance, we calculated the ∆χ^2^ statistic between nested models.

#### 2.3.7. Criterion Validity

To examine criterion validity, we followed the approach of previous research on the FPQ-9 psychometric properties [4,11] and included measures of: (1) chronic pain, (2) average pain intensity, (3) decrease in quality of life due to pain and (4) decrease in quality of life due to anxiety/depression. Considering that fear of pain (as measured by the FPQ-9) is a construct that is independent of pain severity and symptoms of anxiety and depression [4], we examined discriminant validity and expected weak and non-substantive associations between the FPQ-9 and the four selected measures. That is, since fear of pain (e.g., how much the respondent fears painful events) does not necessarily imply that the respondent is currently experiencing pain, chronic pain, or anxiety/depression, these measures were expected to not be associated with the FPQ-9 scores. Due to the ordinal nature of all measures, we report the non-parametric correlation Kendall’s *τ*.

We also computed latent mean differences between groups to evaluate known-groups validity in the case of scalar invariance being established. Since the latent mean is constrained to zero in the reference group, and latent variances are constrained to one in the completely standardised solution, the latent mean differences can be interpreted as effect sizes analogous to Cohen’s (1988) *d* [32]. Additionally, we plotted the latent trait distribution using Kernel density to inform differences between groups.

## 3. Results

The participants’ socio-demographic characteristics are displayed in Table 1. The average age of the participants was 41.8 years (median = 40 years), with the age ranging between 18 to 80 years old, and more than two-thirds of the participants were female (68.2%). Overall, 65% of the participants reported completion of high school; however, 69% were unemployed and/or on benefits, and 73.5% did not have access to a health care card. There were no substantive differences between the original sample and the complete case sample. The distribution of all measures (average pain intensity, chronic pain, quality of life due to anxiety/depression, and quality of life due to pain) for the complete case sample is displayed in Appendix A, Figure A1. Almost one in five participants (19.2%) were experiencing chronic pain, similar to the overall prevalence of chronic pain among Australians aged 45 and over [33]. Furthermore, Appendix A, Figure A1 indicates that while the majority of participants were not experiencing severe pain on the day, tens of participants reported moderate to severe pain according to the measures of average pain intensity (ranging from 0 to 10) and the influence of pain on their quality of life (EQ-5D-5L item; ranging from 1 to 5). For example, 98 participants reported mild to moderate average pain intensity, 48 participants reported that “I have severe pain or discomfort”, and 21 participants reported that “I have extreme pain or discomfort”. As fear of pain is a construct that is independent of pain severity (that is, participants might have a strong fear of pain without been currently experiencing severe pain) [4], we intentionally included participants experiencing mild to severe pain, which is necessary and recommended for questionnaire validation [34].

The sample size calculated to estimate a GGM with nine nodes and edge density of 0.3 reaching a sensitivity of 60% across 80% of all cases was 199 participants. Considering the sample size of 735 participants (complete case sample) in the current study, the study was adequately powered to estimate a GGM with nine nodes and edge density of 0.3 reaching a sensitivity of 60% across 80% of all cases.

The network estimation showed that the FPQ-9 items did not seem to cluster according to the expected dimensions among Indigenous Australian people. As seen from Figure 1, the item “Having a foot doctor remove a wart from your foot with a sharp instrument” was positioned more closely to the items “Getting a papercut on your finger” from the “Fear of Minor Pain” scale and “Breaking your arm” from the “Fear of Severe Pain” scale than to the other “Fear of Medical/Dental Pain” items (Note: Fear of Severe Pain items are coloured orange, Fear of Minor Pain items are coloured blue, and Fear of Medical/Dental Pain items are coloured green). The one-factor model showed poor fit with RMSEA ≥ 0.100.

We then evaluated the fit of the original FPQ-9 structure, which also showed poor fit (Table 2), suggesting that the original FPQ-9 structure is not adequate for Indigenous Australian populations.

We then employed EGA, which indicated a three-dimensional structure (Figure 1, right column), which had a better model fit than the original FPQ-9 structure. The lower TEFIvn value also supported this new three-dimensional structure. Despite being better *relative* to the original FPQ-9 structure, the model fit still was poor and did not achieve acceptable values. For instance, while CFI was good ≥0.900 (recommended value), the RMSEA was poor ≥0.100, which indicates unacceptable model fit. Therefore, we proceeded with the examination of MI and SPEC to investigate if there were issues with individual items. The highest MI and SEPC were from cross-loadings of two items, namely “hotdrink” (MI = 16.065, SEPC = 1.064) and “wartfoot” (MI = 15.601, SEPC = 0.954), on the “Fear of Severe Pain” subscale. Hence, we removed these two items and again performed EGA.

As seen from Figure 2, The EGA on the seven-item FPQ showed: dimension 1, composed of “fallstairs”, “doorhand”, and “breakarm” (orange nodes); dimension 2, composed of “soapeye” and “cutfinger” (blue nodes); and dimension 3, composed of “injmouth” and “injhips” (green nodes). These three dimensions corresponded, respectively, to the theoretical dimensions “Fear of Severe Pain”, “Fear of Minor Pain”, and “Fear of Medical/Dental Pain” from the original FPQ-9 questionnaire.

The evaluation of model fit was acceptable for the EGA FPQ-7 structure (Table 2). The factor loadings and factor correlations of the FPQ-7 were all high (>0.70); however, factor correlations were higher than the desired levels (<0.80) (Table 3). This revised FPQ-9 version with seven items adapted to Indigenous Australians is labelled as the Indigenous Australian Fear of Pain questionnaire (IA-FPQ-7).

After the exclusion of two items, the three-dimensional structure displayed good fit (Table 2). The RMSEA was adequate (<0.100), while CFI was good (≥0.900). The reliability of the “Fear of Severe Pain” scale (Ω = 0.83) and “Fear of Medical/Dental Pain” scale (Ω = 0.80) were good, while the reliability of the “Fear of Minor Pain” scale (Ω = 0.71) was adequate. The IA-FPQ-7 displayed strong CITC among all items considering subscale and total scores, as seen from Figure 3.

The analysis of measurement invariance by age showed adequate fit of the configural model. The progressive comparisons between further constrained models showed that: (1) the constrained thresholds model was not statistically different from the configural model (∆χ^2^ (14) = 10.31; *p* = 0.739); (2) the metric model was not statistically different from the constrained thresholds model (∆χ^2^ (4) = 0.61; *p* = 0.961); and (3) the scalar model was not statistically different from the metric model (∆χ^2^ (4) = 8.96; *p* = 0.083), thus indicating scalar invariance (Table 4). Scalar invariance was achieved, indicating that it is possible to compare the IA-FPQ-7 scores and latent scores across age groups. Similarly, with respect to sex, the configural model achieved adequate fit, and CFI and RMSEA improved as constraints were added, indicating that more parsimonious models adequately described the IA-FPQ-7 item responses across both sexes.

Regarding the analysis of criterion validity, the IA-FPQ-7 subscales and total score displayed weak and non-substantive correlations with average pain intensity, chronic pain, decrease in quality of life due to anxiety/depression, and decrease in quality of life due to pain, as seen in Figure 4.

The comparison of latent scores between groups indicated that fear of severe pain (Mdiff = 0.409; 95% CI [0.212, 0.607]), minor pain (Mdiff = 0.248; 95% CI [0.026, 0.471]), and medical/dental pain (Mdiff = 0.156; 95% CI [−0.047, 0.359]) were higher among participants aged ≥41 years. In addition, fear of severe pain (Mdiff = −0.360; 95% CI [−0.595, −0.125]), minor pain (Mdiff = −0.385; 95% CI [−0.640, −0.129]), and medical/dental pain (Mdiff = −0.324; 95% CI [−0.552, −0.096]) were lower among males. The distribution of fear of pain among these groups is seen in Figure 5.

The same pattern was observed when subscale scores were compared between groups, with scores for fear of severe pain (Mdiff = 1.326; 95% CI [0.782, 1.871]), fear of minor pain (Mdiff = 0.486; 95% CI [0.139, 0.832]), and fear of medical/dental pain (Mdiff = 0.342; 95% CI [−0.062, 0.749]) found higher among participants aged ≥ 41 years. Additionally, fear of severe pain (Mdiff = 1.024; 95% CI [0.429, 1.608]), fear of minor pain (Mdiff = 0.678; 95% CI [0.304, −1.042]), and fear of medical/dental pain (Mdiff = 0.763; 95% CI [0.323, 1.194]) scores were higher among females.

## 4. Discussion

The present study aimed to examine the three-factor structure and validity of the FPQ-9 in a clinical sample of Indigenous Australian people.

Our findings indicated that the original FPQ-9 three-factor structure had a poor fit and did not adequately capture pain-related fear in Indigenous Australian people. Our results are consistent with previous research that showed that Western-developed psychological instruments are not culturally appropriate for Indigenous Australians in their original format [35,36].

We then investigated, using the state-of-the-art methodology Exploratory Graph Analysis, whether an alternative factorial structure (with a different arrangement of items per factor) of the FPQ-9 could be suitable for Indigenous Australian people. However, the alternative factorial structure displayed poor fit and was also inadequate, indicating that the FPQ-9 psychometric problems observed in the Indigenous Australian sample could not be solved solely by rearranging the items into different factors. Upon further examination of the FPQ-9 at an item level, our findings indicated problems with two items: “removal of wart from foot with sharp object” and “gulping down hot drink before it has cooled down. These items “wartfoot” and “hotdrink”, which belong, respectively, to the factors “Fear of Medical/Dental Pain” and “Fear of Minor Pain” in the FPQ-9, displayed cross-loadings on the “Fear of Severe Pain” instead. That is, these items were also measuring “Fear of Severe Pain”, and consequently, the scores from the items “wartfoot” and “hotdrink” cannot be easily interpreted since they are influenced by more than one factor. Since cross-loadings lead to a non-interpretable item score, cross-loadings are considered a threat to the construct validity [37]. For this reason, we removed the two items from the questionnaire.

Moreover, this decision to remove these items was also guided by the knowledge that Indigenous Australian people may look for alternative practices, for instance, seeking a traditional healer, rather than actively seeking modern medicine for removing a wart from their foot. Culturally appropriate interventions to deal with pain are preferred by Indigenous Australian people [38]. Similarly, there is a strong possibility that Indigenous Australian people reported lower pain-related fear associated with gulping a hot drink on the background of more frequent exposure to trauma-induced injuries and falls [39,40].

After the removal of the two items, the three-dimensional structure of the new IA-FPQ-7 indicated by EGA displayed a good fit, and the three-factor structure corresponded to the original FPQ-9 structure (“Fear of Severe Pain”, “Fear of Minor Pain”, and “Fear of Medical/Dental Pain”). The IA-FPQ-7 items were good measures of the construct; however, the factor correlations were higher than the adequate levels (>0.80), positing threats to discriminant validity. The IA-FPQ-7 structure also showed good reliability for the “Fear of Severe Pain” and “Fear of Medical/Dental Pain” subscales, while the reliability of the “Fear of Minor Pain” subscale was adequate.

The IA-FPQ-7 displayed measurement invariance across gender and age groups, indicating that the same seven items had the same meaning across groups (i.e., no differential item functioning), and direct comparison between groups (e.g., fear of pain among men compared to women) can be conducted. The IA-FPQ-7 subscale and total scores showed weak and non-significant correlations with measures of chronic pain, average pain intensity, and decrease of quality of life due to pain or anxiety/depression. This weak, non-substantive association is consistent with a robust literature [7,14]. It also strengthens the specificity of the fear of pain as a construct independent of pain severity [14] and symptoms of anxiety and depression. The comparison of latent scores indicated that pain-related fear of severe pain, fear of minor pain, and fear of medical/dental pain was higher among female participants and those above the ages of 41 years [4].

Contrary to previous findings [41], our findings indicate higher fear of medical/dental pain among older participants, which needs to be interpreted in the socio-historical context of Indigenous Australian participants. Indigenous people have been exposed to several major catastrophic events, including loss of culture and land, forced removal of children, missing generations [21,42], along with systemic racism, leaving many with distrust and ambivalent feelings (such as fear) towards medical professionals and the medical health care system [43].

Research shows that these devastating experiences of trauma continue to be passed down as intergenerational trauma through local folklore, child-raising practices, childhood experiences, and especially drug abuse, forming a vicious cycle [39]. In line with existing literature, women in our study also reported more fear of pain than men [44]. It is likely that their experience of higher levels of domestic violence, trauma, and abuse [40,41] may also contribute to higher reports of pain-related fear.

The current study has several strengths and limitations. Strengths include that the study followed recommended practices for cultural adaptation of psychological instruments and received oversight from an Indigenous Reference Group [17,20]; secondly, the study has a large sample (*n* = 750) considering the recruitment difficulties concerning Indigenous populations [45,46], with recruitment conducted over large distances (travelling 700 km to the west of the city of Adelaide, the capital of the State of South Australia; 400 km east and 800 km north); and lastly, we have used modern psychometric methods to provide the best evidence about the IA-FPQ-7 psychometric properties [21,47]. One potential limitation could be the representativeness of the sample. Despite non-probability purposive sampling being employed, the study sample was reasonably representative of the Indigenous Australian population [17] from South Australia. Moreover, while representativeness can be desirable for certain studies (e.g., studies describing the prevalence of fear of pain in the population), a non-representative sample does not entail that the item parameters are biased [4,48] or necessarily limits the generalisability of the findings [48,49].

In summary, the evaluation of criterion validity with respect to theoretically unrelated variables (e.g., chronic pain, pain intensity) and know-groups comparison (e.g., higher fear of pain among women) was consistent with theoretical expectation and previous empirical research, providing further evidence to support the construct validity of the IA-FPQ-7 among Indigenous Australians.

## 5. Conclusions

In conclusion, our findings showed that, after the exclusion of two cross-loading items, an adapted version IA-FPQ-7 displayed evidence of construct validity and adequate reliability for the assessment of fear of pain in a sample of Indigenous Australian people. The IA-FPQ-7 scale could be used to better understand what role pain-related fear might play in recovery, disability, and distress associated with chronic pain in Indigenous Australian populations. This could allow for more tailored and timely interventions for managing pain in Indigenous Australian communities. Future research should further examine the IA-FPQ-7 psychometric properties in Indigenous Australian population from other states and investigate other aspects of construct validity, such as convergent validity and predictive validity.

## Figures and Tables

**Figure 1 ijerph-19-06256-f001:**
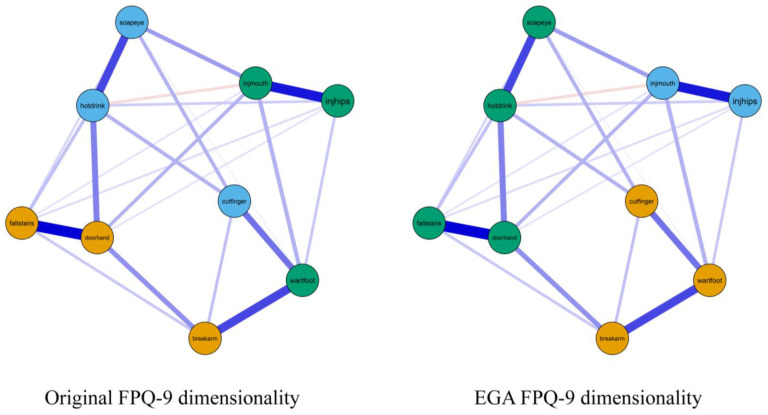
Networks of the Fear of Pain Questionnaire-9. Note: On the right graph, nodes were coloured according to the FPQ-9 theoretical dimensions of Fear of Medical/Dental Pain (**green**), Fear of Minor Pain (**blue**), and Fear of Severe Pain (**orange**). On the left graph, nodes were coloured according to the EGA-identified dimensions. Positive edges are displayed as blue lines, and negative edges are displayed as red lines. Edge weights are represented by the thickness and saturation of the edges.

**Figure 2 ijerph-19-06256-f002:**
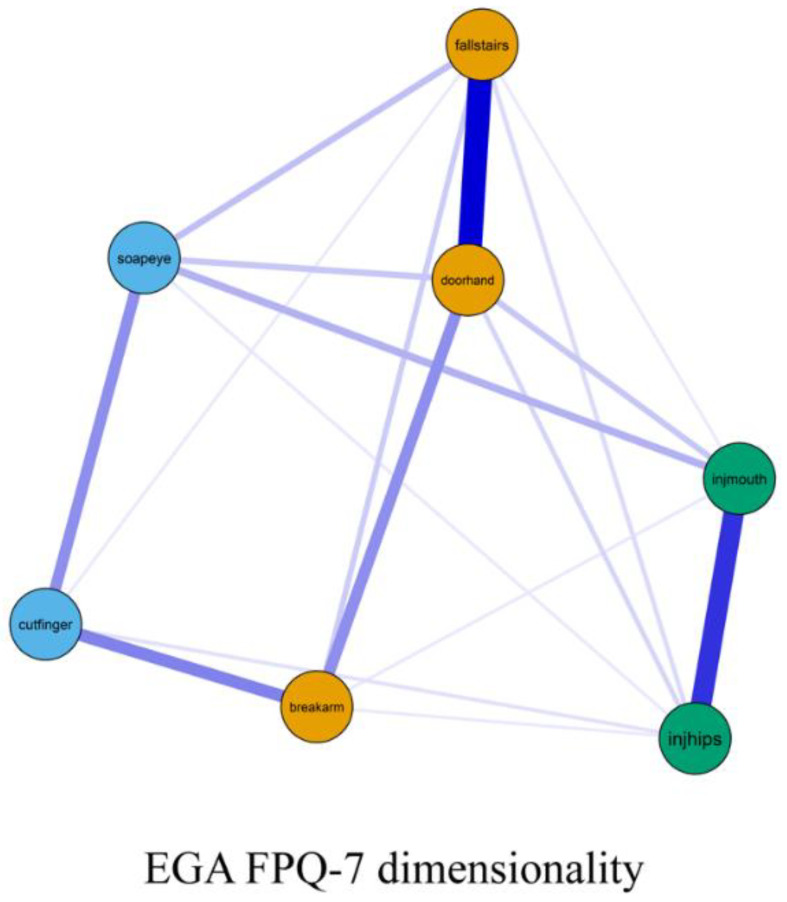
Network of the Fear of Pain Questionnaire-7 (IA-FPQ-7). Note: Nodes were coloured according to the EGA-identified dimensions, which corresponded to the theoretical dimensions of Fear of Medical/Dental Pain (**green**), Fear of Minor Pain (**blue**), and Fear of Severe Pain (**orange**). Positive edges are displayed as blue lines, and negative edges are displayed as red lines. Edge weights are represented by the thickness and saturation of the edges.

**Figure 3 ijerph-19-06256-f003:**
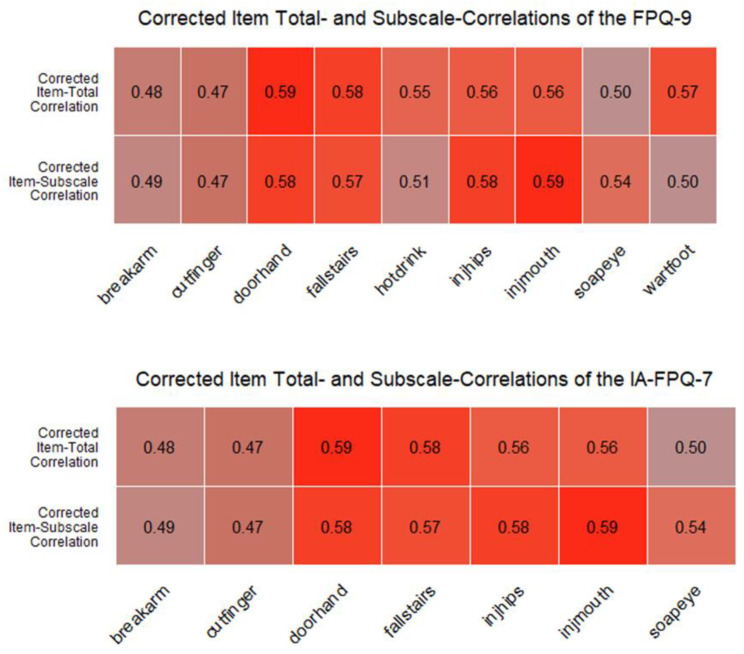
Corrected item total correlations of the FPQ-9 and IA-FPQ-7. Note: The saturation of the colour indicates the strength of the correlation (i.e., stronger shades of red indicate stronger correlations).

**Figure 4 ijerph-19-06256-f004:**
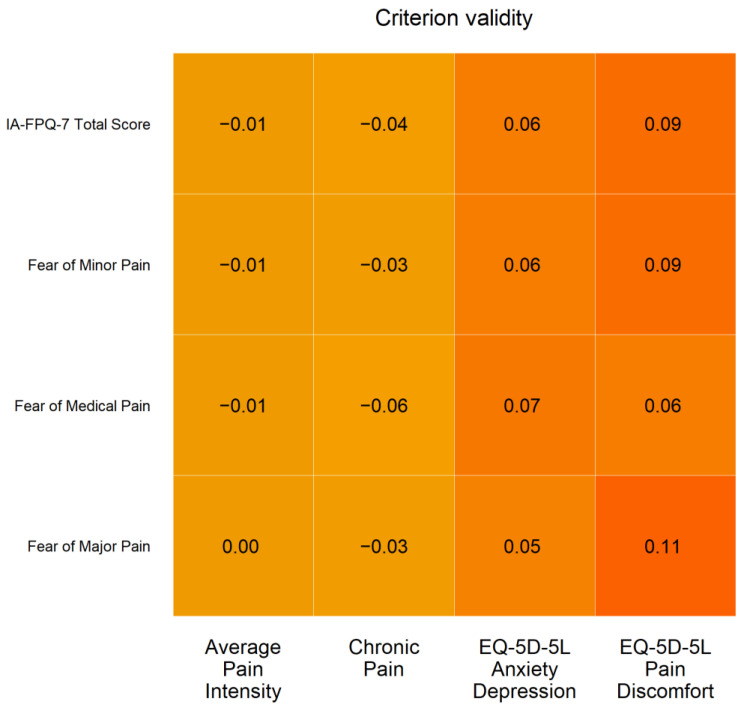
Criterion validity of the IA-FPQ-7. Note: The saturation of the colour indicates the strength of the correlation (i.e., stronger shades of orange indicate stronger correlations).

**Figure 5 ijerph-19-06256-f005:**
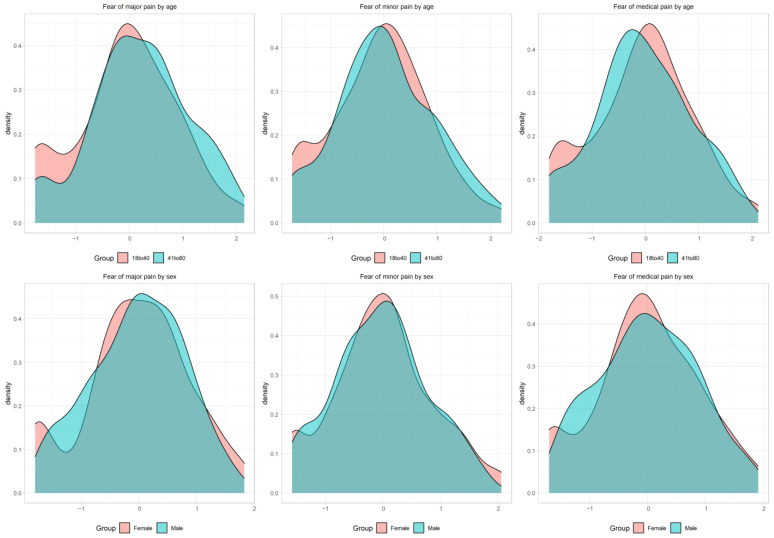
Latent scores density plots of fear of pain by age and sex. Note: The Kernel density plots indicate the distribution of factor scores according to age and sex. The *x*-axis denotes the factors scores, and the *y*-axis denotes the kernel estimates of the probability density.

**Table 1 ijerph-19-06256-t001:** Characteristics of study participants.

	HPOVC Study (*n* = 750)		Complete Case Sample (*n* = 735)	
	*N*	%	*N*	%
**Age**				
Mean (SD)	40.7 (14.6)		40.7 (14.5)	
Missing	0	0.0	0	0.0
**Sex**				
Male	234	31.2	228	31.0
Female	508	67.7	501	68.2
Missing	8	1.1	6	0.8
**Education**				
High School completed	487	64.9	480	65.3
Technical education or university	253	33.8	246	33.5
Missing	10	1.3	9	1.2
**Employment**				
Employed	193	25.8	189	25.7
Unemployed/Benefits	517	68.9	507	69.0
Other/Missing	40	5.3	39	5.3
**Access to health care card**				
Yes	161	21.5	157	21.3
No	551	73.5	540	73.5
Do not know/Missing	38	5.0	38	5.2

**Table 2 ijerph-19-06256-t002:** Model fit comparison of the original FPQ structure and EGA identified dimensions.

	χ^2^	*df*	*p*-Value	RMSEA	90% CI	CFI	TEFIvn
One-factor structure	311.694	27	<0.001	0.143	(0.129–0.158)	0.957	0.000
Original FPQ-9 structure	263.929	24	<0.001	0.139	(0.125–0.155)	0.964	−1.845
EGA FPQ-9 structure	174.880	24	<0.001	0.111	(0.096–0.126)	0.977	−2.077
EGA FPQ-7 structure ^a^	37.408	11	<0.001	0.068	(0.045–0.093)	0.981	-

Note: ^a^ The one-factor, original FPQ-9 and EGA FPQ-9 structures displayed in the first three rows were fitted to nine FPQ items. The EGA FPQ-7 structure displayed in the fourth row was fitted to seven FPQ items. Since the TEFIvn provides only a measure of relative fit of the model compared to other models fitted to the same data (i.e., same number of items), the TEFIvn was omitted for the EGA FPQ-7 structure.

**Table 3 ijerph-19-06256-t003:** Factor model of three-factor IA-FPQ-7 in Indigenous Australian.

Items	Factor Loadings		
	Fear of Major Pain	Fear of Minor Pain	Fear of Medical/Dental Pain
Falling down a flight of concrete stairs	**0.87 (0.02)**	0.00 (0.00)	0.00 (0.00)
2. Having someone slam a heavy car door on your hand	**0.88 (0.02)**	0.00 (0.00)	0.00 (0.00)
3. Breaking your arm	**0.73 (0.03)**	0.00 (0.00)	0.00 (0.00)
4. Getting strong soap in both your eyes while bathing or showering	0.00 (0.00)	**0.83 (0.03)**	0.00 (0.00)
5. Getting a papercut on your finger	0.00 (0.00)	**0.77 (0.03)**	0.00 (0.00)
6. Receiving an injection in your mouth	0.00 (0.00)	0.00 (0.00)	**0.85 (0.02)**
7. Receiving an injection in your hip/buttocks	0.00 (0.00)	0.00 (0.00)	**0.87 (0.02)**
**Factor Correlations**			
Fear of Major Pain × Fear of Minor Pain	0.80 (0.03)		
2. Fear of Major/Pain × Fear of Medical/Dental Pain	0.86 (0.02)		
3. Fear of Minor Pain × Fear of Medical/Dental Pain	0.84 (0.03)		

Note: Table reports estimates and standard errors (Estimates (SE)). The factor loadings on the theoretical FPQ factors are highlighted in bold.

**Table 4 ijerph-19-06256-t004:** Measurement invariance according to sex and age.

Model	χ^2^	*df*	*p-*Value	RMSEA	90% CI	CFI	∆χ^2^ (*df*)	*p-*Value
**Age**								
Configural	53.082	22	<0.001	0.074	(0.049, 0.100)	0.994	-	-
Constrained Thresholds	69.086	36	0.001	0.060	(0.038, 0.081)	0.993	10.31 (14)	0.739
Metric	65.331	40	0.007	0.050	(0.026, 0.071)	0.995	0.61 (4)	0.961
Scalar	77.425	44	0.001	0.054	(0.038, 0.081)	0.993	8.96 (4)	0.062
**Sex**								
Configural	76.879	22	<0.001	0.099	(0.075, 0.124)	0.989	-	-
Constrained Thresholds	93.578	36	<0.001	0.079	(0.060, 0.099)	0.988	10.00 (14)	0.762
Metric	90.535	40	<0.001	0.070	(0.051, 0.090)	0.989	4.13 (4)	0.389
Scalar	85.656	44	<0.001	0.061	(0.041, 0.080)	0.991	3.47 (4)	0.482

## Data Availability

Data made available upon request.

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
