# Peer review of "Assessment of Pain-Related Fear in Indigenous Australian Populations Using the Fear of Pain Questionnaire-9 (FPQ-9)"

_ijerph, 2022, doi:10.3390/ijerph19106256_

Round 1
Reviewer 1 Report
This study expands our understanding and validity of the FPQ-9 by focusing on autochthonous populations in Australia. The study is relevant, although, the following aspects need to be clarified:
-Provide the age range of participants tested.
-Why there is predominance of female participants in the sample?
-Were participants tested free of depression or anxiety (unrelated to pain)? How did the authors evaluate this? I think this is crucial to understand better the reported results.
-Did participants receive any financial remuneration for participation?
-Were the tests presented to participants in a native language other than English? please clarify.
-A discussion about the scores obtained need to be included.
Author Response
Reviewer 1
This study expands our understanding and validity of the FPQ-9 by focusing on autochthonous populations in Australia. The study is relevant, although, the following aspects need to be clarified:
-Provide the age range of participants tested.
Author response: We thank the reviewer for their feedback. We have now included the participants’ age range in the Results section (Line 187-188).
-Why there is predominance of female participants in the sample?
Author response: We agree that there is predominance of female participants which is purely coincidental.
-Were participants tested free of depression or anxiety (unrelated to pain)? How did the authors evaluate this? I think this is crucial to understand better the reported results.
Author response: We thank the reviewer for their feedback. As described in the methods section the EQ-5D-5L measure of health-related quality of life was used. The EQ-5D-5L, however, is not a diagnostic measure of depression and anxiety. The scores of the EQ-5D-5L item measuring anxiety/depression are displayed in Supplementary Appendix A Figure 1 (second row, first column) indicating that the majority of participants (n=314) endorsed the category stating that they were 'not anxious or depressed'''
-Did participants receive any financial remuneration for participation?
Author response: We thank the reviewer for their feedback. We have added below information (Line 82-83) in Design & setting section.
“For their time each participant received reimbursement of $50 voucher at baseline, 12-month and 24-month follow-ups.”
-Were the tests presented to participants in a native language other than English? please clarify.
Author response: The tests were only administered in English as English is spoken by an estimated 80% of Aboriginal and Torres Strait Islander people, and is the first and only language spoken by many Aboriginal children
-A discussion about the scores obtained need to be included.
Author response: We thank the reviewer for their feedback. We have included discussion about the scores relevant to the aims in the discussion section. Additional scores have been displayed in Figure 4 (revised main manuscript), we reported the correlation of the IA-FPQ-7 total and subscales scores with the EQ-5D-5L item measuring anxiety/depression (third column). Figure 5 (revised main manuscript) displays the IA-FPQ-7 latent scores by sex and age.
Reviewer 2 Report
I read with great interest the article titled “Assessment of pain related fear in Indigenous Australian 2 population using the Fear of Pain-9 Questionnaire (FPQ-9)” in which the authors concluded that an adapted version IA-FPQ-7 displayed evidence of construct validity and adequate reliability for the assessment of fear of pain in a sample of Indigenous Australian people. This could allow for more tailored and timely interventions for managing pain in Indigenous Australian communities, since optimal assessment is the first step towards improving pain care for Indigenous communities. The article is well structured, well written, and presented the topic in an interesting way.
Comments and suggestions:
- Line 83: “For further information on the HPV study, please refer to [17].” The statement can be modified. For example, the reference 17 can be moved to be a reference for lines 67- 70 (first sentence in the methodology section), since you referred to the term “HPV study” before mentioning this reference.
- Were the inclusion criteria for this study the same as the HPV study? Or did you exclude patients who had certain factors that might interfere with their perception of pain or fear of pain?
- Several abbreviations were mentioned without them being previously defined. Please mention the full term upon its first appearance in the manuscript prior to the utilization of the abbreviation.
- A sample of 1,011 Aboriginal adults was recruited at baseline and 750 participated in the 12-months follow-up (74% retention rate; the study was suspended early due to the COVID-19 pandemic).
Can you please calculate the power of the study, and mention it here for the readers? Despite mentioning the representativeness of the sample in your limitations, mentioning the study statistical power might add strength to the article. - Line 188: “similar to the overall prevalence of chronic pain among Australians aged 45 and over [32] (AIHW, 2020).”
Preferably, move comparisons with previous studies to the discussion. - Line 368: “intergenerational trauma2 through local folklore”
remove the “2” if not needed in the sentence. - Please add definition of what does X axis and Y axis represent in each of the presented figures.
Author Response
Reviewer 2
- Line 83: “For further information on the HPV study, please refer to [17].” The statement can be modified. For example, the reference 17 can be moved to be a reference for lines 67- 70 (first sentence in the methodology section), since you referred to the term “HPV study” before mentioning this reference.
Author response: We thank the reviewer for this feedback. The sentence has been amended.
- Were the inclusion criteria for this study the same as the HPV study? Or did you exclude patients who had certain factors that might interfere with their perception of pain or fear of pain?
Author response: Similar inclusion criteria was used for this study.
- Several abbreviations were mentioned without them being previously defined. Please mention the full term upon its first appearance in the manuscript prior to the utilization of the abbreviation.
Author response: We thank the reviewer for their feedback. We hereby confirm that we have checked and have added full term prior to first appearance.
- A sample of 1,011 Aboriginal adults was recruited at baseline and 750 participated in the 12-months follow-up (74% retention rate; the study was suspended early due to the COVID-19 pandemic).
Can you please calculate the power of the study, and mention it here for the readers? Despite mentioning the representativeness of the sample in your limitations, mentioning the study statistical power might add strength to the article.
Author response: We have now described how we conducted the power analysis for this study in the Methods (Line 120-125) and Results (Line 212-216) sections.
- Line 188: “similar to the overall prevalence of chronic pain among Australians aged 45 and over [32] (AIHW, 2020).”
Preferably, move comparisons with previous studies to the discussion.
Author response: We thank the reviewer for their comment, but this line provides context to the reader for follow up sections and hence has been retained.
- Line 368: “intergenerational trauma2 through local folklore”
remove the “2” if not needed in the sentence.
Author response: We thank the reviewer for their feedback. We hereby confirm that we have amended the sentence.
- Please add definition of what does X axis and Y axis represent in each of the presented figures.
Author response: We thank the reviewer for their feedback. We have now included the description of what the X and Y axis represent in Figure 5 (Line 312-313).
This manuscript is a resubmission of an earlier submission. The following is a list of the peer review reports and author responses from that submission.